# MOLECULE OPTIMIZATION BY EXPLAINABLE EVOLUTION

*Binghong Chen[1], *Tianzhe Wang[1,5], Chengtao Li[2], Hanjun Dai[3], Le Song[4]
[1]Georgia Institute of Technology    [2]Galixir    [3]Google Research, Brain Team
[4]Mohamed bin Zayed University of AI    [5]Shanghai Qi Zhi Institute

## ABSTRACT

Optimizing molecules for desired properties is a fundamental yet challenging task in chemistry, material science, and drug discovery. This paper develops a novel algorithm for optimizing molecular properties via an Expectation-Maximization (EM) like explainable evolutionary process. The algorithm is designed to mimic human experts in the process of searching for desirable molecules and alternate between two stages: the first stage on explainable local search which identifies rationales, *i.e.*, critical subgraph patterns accounting for desired molecular properties, and the second stage on molecule completion which explores the larger space of molecules containing good rationales. We test our approach against various baselines on a real-world multi-property optimization task where each method is given the same number of queries to the property oracle. We show that our evolution-by-explanation algorithm is 79% better than the best baseline in terms of a generic metric combining aspects such as success rate, novelty, and diversity. Human expert evaluation on optimized molecules shows that 60% of top molecules obtained from our methods are deemed successful.

## 1 INTRODUCTION

The space of organic molecules is vast, the size of which is exceeding $10^{60}$ (Reymond et al., 2010). Searching over this vast space for molecules of interest is a challenging task in chemistry, material science, and drug discovery, especially given that molecules are desired to meet multiple criteria, *e.g.*, high potency and low toxicity in drug discovery. When human experts optimize molecules for better molecular properties, they will first come up with rationales within desirable molecules. Typically, the rationales are subgraphs in a molecule deemed to contribute primarily to certain desired molecular properties. Once rationales are identified, chemists will design new molecules on top of rationales hoping that, the desired properties of new molecules will be further enhanced due to the existence of rationale and changes of non-rationale parts. The cycle of identifying molecular rationales and redesigning new hypothetical molecules will be carried on until molecules that meet certain property criteria are discovered.

In this paper, we develop a novel algorithm that mimics the process of molecule optimization by human experts. Our algorithm finds new molecules with better properties via an EM-like explainable evolutionary process (Figure 1). The algorithm alternates between two stages. During the first stage, we use an explainable local search method to identify rationales within high-quality molecules that account for their high property scores. During the second stage, we use a conditional generative model to explore the larger space of molecules containing useful rationales.

Our method is novel in that we are using explainable models to help us exploit useful patterns in the molecules, yet leveraging generative models to help us explore the molecule landscape. Comparing to existing methods that directly learn a generative model using Reinforcement Learning or perform continuous optimization in the latent space of molecules (Olivecrona et al., 2017; You et al., 2018a; Dai et al., 2018b), our method is more sample-efficient and can generate more novel and unique molecules that meet the criteria.

We evaluate our algorithm against several state-of-the-art methods on a molecule optimization task involving multiple properties. Compared with baselines, our algorithm is able to increase the success

---

*Correspondence to: Binghong Chen <binghong@gatech.edu>. * indicates equal contribution.
Source code at `https://github.com/binghong-ml/MolEvol`.

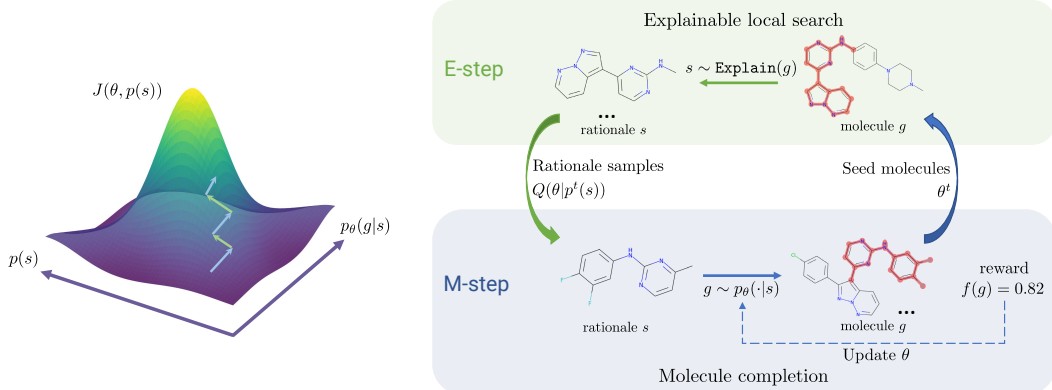

Figure 1: Overview of our EM-like evolution-by-explanation algorithm. Left: climbing up the energy landscape $J(\theta, p(s))$ by alternatively taking an E-step and M-step. Right: illustrations for the E-step and M-step. In the E-step of time $t$, we draw samples from $Q(p(s)|\theta^{t-1})$ to approximate $Q(\theta|p^t(s))$ using rationales extracted from the seed molecules via an explainable model. Then in the M-step, we optimize $Q(\theta|p^t(s))$ w.r.t. $\theta$, *i.e.* pushing the graph completion model $p_\theta(\cdot|s)$ towards generating higher scoring molecules conditioned on the rationale samples.

rate by 50%, novelty by 14%, while having a competitive diversity. We further propose a new metric, QNU score, to jointly consider all three aspects, and show that we achieve a score of 52.7% compared with 29.5% by the best baseline. We also ask experienced chemists to evaluate top-50 generated molecules and find that 30 of them are as good as existing ones.

The main contributions of this paper are summarized below:

- We propose a novel EM-like evolution-by-explanation algorithm for molecule optimization;
- We present a novel, principled, explainable graph model based on an information-theoretic approach to extract subgraphs essential for maintaining certain desired properties;
- Our approach outperforms existing state-of-the-arts by a large margin in terms of success rate (50% better), novelty (14% better), and an overall metric (79% better) on a real-world multi-property optimization task.

## 2 RELATED WORK

There has been a surge of interest in using machine learning to discover novel molecules with certain properties in recent years. Most of the existing work defines a generative model for either the SMILES strings (Weininger, 1988) or molecular graphs, and uses Reinforcement Learning algorithms to optimize the properties of the generated molecules (Segler et al., 2018; Olivecrona et al., 2017; Guimaraes et al., 2017; You et al., 2018a; Popova et al., 2018; 2019; Samanta et al., 2019; Zhou et al., 2019; De Cao & Kipf, 2018; Kearnes et al., 2019; Shi et al., 2020; Jin et al., 2020). Others optimize the continuous representation of molecules in a latent space learned by variants of variational autoencoders (Kusner et al., 2017; Dai et al., 2018b; Jin et al., 2018; Gómez-Bombarelli et al., 2018; Kang & Cho, 2018; Liu et al., 2018; Kajino, 2019). More recent work attempts Evolutionary algorithms (Nigam et al., 2020; Leguy et al., 2020; Winter et al., 2019), or focuses on finding high-quality molecules with synthesis paths (Bradshaw et al., 2019; Korovina et al., 2020; Gottipati et al., 2020). Most similar to our approach is RationaleRL (Jin et al., 2020), which extracts subgraphs from seed molecules using Monte Carlo Tree Search (MCTS) and generates full molecules by completing the subgraphs. Compared with previous work, our approach is the first to incorporate an explainable model in the iterative search process.

Existing work on explainable models approaches the problems from three directions. The first line of work uses gradients of the outputs with respect to inputs to identify the salient features in the inputs (Simonyan et al., 2013; Springenberg et al., 2014; Baehrens et al., 2010); the second line of work approximates the model with simple interpretable models, such as locally additive mod-

els (Bach et al., 2015; Kindermans et al., 2016; Ribeiro et al., 2016; Lundberg & Lee, 2017; Shrikumar et al., 2017); the third line of work defines input pattern selection operators, such that the outputs of the model based on the selected input patterns have high mutual information with the original model outputs (Chen et al., 2018; Ying et al., 2019). Our explainable model is different from GNNExplainer (Ying et al., 2019) in that we optimize the discrete subgraph structure with learned variational predictor, instead of directly feeding continuous edge masking into the target model.

## 3 PROBLEM SETTING

In this paper, we study the problem of discovering molecules $g$ from the molecular space $\mathcal{G}$ with a high property score, measured by a scoring function $f$. And usually, there is a set of seed molecules $\mathcal{G}_0 \subset \mathcal{G}$ from experts with high scores to start with. More formally, the problem can be stated as

**Molecule Optimization.** Given a scoring function $f : \mathcal{G} \mapsto [0, 1]$, and a set of seed molecules $\mathcal{G}_0 \subset \mathcal{G}$, the goal is to learn a molecule generative model $p(g)$ such that the expected score of the generated molecules is maximized, *i.e.*,

$$\max_{p(\cdot)} \ \mathbb{E}_{g \sim p(\cdot)}[f(g)] = \int_{g \in \mathcal{G}} p(g) f(g) dg \tag{1}$$

To prevent the model $p(g)$ from generating a small set of fixed molecules with high scores, we additionally require the learned distribution to be both *novel* and *diverse*, *i.e.*, generating molecules that are dissimilar to the set of reference molecules (a subset of $\mathcal{G}_0$) and each other.

The molecule optimization problem in Eq (1) is combinatorial in nature, which poses a significant challenge. To mimic the scientific discovery process, we allow the algorithm to query $f$ on new molecules under a querying budget. Examples of some well-known scoring functions include the QED score measuring the drug-likeness (Bickerton et al., 2012), the SA score measuring the synthetic accessibility (Ertl & Schuffenhauer, 2009), the TPSA score measuring the ability to permeate cells (Prasanna & Doerksen, 2009), etc. The scoring function is general and could also encode multi-property objectives (Olivecrona et al., 2017; Brown et al., 2019). Optimizing multiple properties together suffers from the sparsity of high scores, a scenario which is shown to be more challenging than single property optimization (Jin et al., 2020).

When experts are optimizing the molecular property, they will first look for substructures that result in the formation of that property, and use them as the foundation for building novel molecules. These subgraphs are called rationales (examples in Figure 1). The set of rationales is formally defined as,

$$\mathcal{S} = \{s \mid \exists g \in \mathcal{G}, \quad \text{s.t.} \ s \text{ is a subgraph of } g\}. \tag{2}$$

## 4 OUR FRAMEWORK

Our novel framework for optimizing molecular property with generative models consists of a modeling component and an algorithm component. In our modeling component, we propose a rationale-based hierarchical generative model for $p(g)$, which first generates rationales and then completes molecules. In our algorithm component, we design an alternating optimization procedure that interleaves between rationale distribution optimization and molecule generative model optimization. Furthermore, we develop a novel explainable graph model to effectively carry out the rationale model optimization. Next, we will first start describing our hierarchical generative model.

### 4.1 RATIONALE-BASED HIERARCHICAL GENERATIVE MODEL

To tackle the challenging search problem, we develop a hierarchical generative model that mimics the process of molecule optimization by human experts. In our model, we first sample rationales $s$ from a distribution $p(s)$, and then molecules $g$ will be generated according to conditional distribution $p_\theta(g|s)$. More specifically, our overall molecular generative model $p_\theta(g)$ can be defined as

$$p_\theta(g) = \int_{s \in \mathcal{S}} p(s) \, p_\theta(g|s) \, ds, \tag{3}$$

where $\theta$ is the parameter of the conditional generative model, $p(s)$ is the latent rationales distribution.

Here $p_\theta(g|s)$ is a graph completion model from rationale $s$. The architecture of $p_\theta(g|s)$ can be arbitrary. In this work, we use a latent variable model with a Gaussian prior $p(z)$,

$$p_\theta(g|s) = \int_z p(z)p_\theta(g|s,z)dz, \tag{4}$$

where $p_\theta(g|s,z)$ is a variant of the GraphRNN (You et al., 2018b; Liu et al., 2018) by conditioning the graph generation on subgraphs. As part of the initialization, $p_\theta(g|s)$ is first pretrained on ChEMBL (Gaulton et al., 2017), a drug-like molecule dataset, in the same fashion as the variational autoencoder (Kingma & Welling, 2013), where the encoder is a standard GCN with atoms as vertices and bonds as edges.

Note that different from $p(z)$, which is a fixed prior, $p(s)$ will be updated in each round. And since representing a distribution on $\mathcal{S}$ is difficult, we will use particles to represent $p(s)$ in the algorithm.

In order to improve the diversity of the generated molecules, we will also regularize the entropy of the rationale distribution $p(s)$, leading to the following diversity-promoting objective function

$$J(\theta, p(s)) = \mathbb{E}_{g \sim p_\theta(\cdot)}[f(g)] + \lambda \cdot \mathbb{H}[p(s)], \tag{5}$$

with a hyperparameter $\lambda > 0$ controlling the strength of the regularization.

## 4.2 ALTERNATING OPTIMIZATION ALGORITHM

As the rationales and the molecules are coupled in the molecular graph space, directly optimizing the diversity-promoting objective in Eq (5) would be challenging. Therefore we seek to optimize $p_\theta(g|s)$ and $p(s)$ in an alternating fashion, akin to the Expectation-Maximization (EM) algorithm. That is, the algorithm alternates between two stages:

- **Expectation step** (E-step) for obtaining an updated distribution $p(s)$, and
- **Maximization step** (M-step) for improving the molecule completion model $p_\theta(g|s)$.

We name this algorithm `MolEvol` (Algorithm 1) by making an analogy to evolving a group of molecules over time (Figure 1). Assume that, at iteration $t-1$, we already have $p_{\theta^{t-1}}(g|s)$ and $p^{t-1}(s)$, and the set of seed samples $\mathcal{G}^{t-1}$ drawn from $p_{\theta^{t-1}}(g|s)$. Then, at iteration $t$, we have,

**E-step.** We want to maximize the objective $J$ with respect to the latent distribution $p(s)$ given $p_{\theta^{t-1}}(g|s)$. That is

$$\max_{p(s)} Q(p(s)|\theta^{t-1}) := \int_{s \in \mathcal{S}} p(s)\Big(\int_{g \in \mathcal{G}} p_{\theta^{t-1}}(g|s)f(g)dg\Big)ds - \lambda \int_{s \in \mathcal{S}} p(s)\log p(s)ds. \tag{6}$$

which is a maximum entropy estimation problem. Interestingly, the solution of the above optimization problem can be obtained in close form.

$$p^t(s) = \operatorname*{argmax}_{p(s)} Q(p(s)|\theta^{t-1}) = \frac{1}{Z_\theta} \exp\Big(\frac{1}{\lambda}\mathbb{E}_{g \sim p_{\theta^{t-1}}(\cdot|s)}[f(g)]\Big), \tag{7}$$

where $Z_\theta$ is a normalizing constant. This updated distribution for the latent rationales will be needed for the later M-step. However, since directly integrating with respect to $p^t(s)$ is difficult, we will leverage sampling strategies and obtain $m$ particles $\{s_i\}_{i=1}^m$ from this distribution for later use in M-step. However, computing the normalizing constant $Z_\theta$ is difficult, making direct sampling from $p^t(s)$ not straightforward. Standard sampling algorithms like Markov Chain Monte Carlo (Andrieu et al., 2003) could be extremely slow due to the lack of a good proposal distribution and the absence of gradients in the discrete graph space.

To address this challenge, we will maintain a finite support set $\mathcal{S}^t$ as the proposal, which is obtained from an explainable graph model (more details in the next section). More specifically, suppose the explainable graph model, `Explain`$(\cdot) : \mathcal{G} \mapsto \mathcal{S}$, can take a graph input $g$ and output the corresponding rationale $s$ which explains why the graph $g$ can obtain a high property score according to $f(g)$. Then support set $\mathcal{S}^t$ can be maintained as follows

$$\mathcal{S}^t = \bigcup_{i=1}^t \Big\{ \texttt{Explain}(g) : g \in \mathcal{G}^i \Big\}, \tag{8}$$

where $\mathcal{G}^0$ is provided to the algorithm initially by experts. The rationales $s \in \mathcal{S}^t$ will be treated as the set of particle locations for representing $p^t(s)$. Furthermore, for each of these particle locations, we will compute its unnormalized probability according to $p^t(s)$ in Eq (7), and then re-sample a set of $m$ particles, $\{s_i\}_{i=1}^m$, as the final representation for $p^t(s)$ (Andrieu et al., 2003).

**M-step.** With $\{s_i\}_{i=1}^m$ from $p^t(s)$, the Monte Carlo estimate of the objective function in Eq (5) becomes

$$Q(\theta|p^t(s)) \approx \sum_{i=1}^m \int p_\theta(g|s_i) f(g) dg + \text{constant.} \tag{9}$$

We can then maximize it with respect to the parameters $\theta$ using REINFORCE,

$$\theta^t \leftarrow \theta^{t-1} + \alpha \frac{1}{m} \sum_{i=1}^m f(g_i) \nabla \log p_{\theta^{t-1}}(g_i|s_i), \quad \text{where } \alpha > 0, \ g_i \sim p_{\theta^{t-1}}(\cdot|s_i). \tag{10}$$

After the parameter is updated to $\theta^t$, we will sample a seed set of molecules $\mathcal{G}^t$ from $p_{\theta^t}(g|s)$ by completing the rationale samples $\{s_i\}_{i=1}^m$ using the updated model. That is

$$\mathcal{G}^t = \{g_i\}_{i=1}^{n_s}, \quad \text{where } g_i \sim g_{\theta^t}(\cdot|s), s \sim \text{Uniform}(\{s_1, s_2, \ldots, s_m\}). \tag{11}$$

The overall algorithm is summarized in Algorithm 1. $p(s)$ and $p_\theta(g|s)$ are updated in the E-step (line 3-4) and M-step (line 5-8), respectively. A discussion on its convergence can be found in Appendix A.2.

---

**Algorithm 1:** Molecule Optimization by Explainable Evolution (`MolEvol`)

---

**Input:** Seed molecules $\mathcal{G}^0$, pretrained graph completion model $p_\theta(g|s)$ on ChEMBL.

1  Initialize $S^0 = \{\}$.
2  **for** $t \leftarrow 1$ **to** $N_{rounds}$ **do**
3      $\mathcal{S}^t = \mathcal{S}^{t-1} \cup \{\text{Explain}(g) : g \in \mathcal{G}^{t-1}\}$.
4      Sample $s_1, s_2, \cdots, s_m$ from $\mathcal{S}^t$ using Eq (7) with self-normalization.
5      **for** $j \leftarrow 1$ **to** $N_{epochs}$ **do**
6          Sample $g_1, \cdots, g_m$ from $p_\theta(g|s_1), \cdots, p_\theta(g|s_m)$ respectively.
7          Update $\theta$ with REINFORCE (Eq (10)).
8      Sample seed molecules $\mathcal{G}_t$ with Eq (11).
9  **return** $p_\theta(g)$

---

### 4.3 Explainable Graph Model for Rationales

In the E-step, it is crucial to update the support set for the rationales, such that the particles can be placed in spaces where $p^t(s)$ is large. As we optimize $p_{\theta^t}(g|s)$, this model can generate molecules with improved property scores. Intuitively, we would also like to have an increasingly "good" support set for the rationales. To do this, we will identify substructures in the current seed set of molecules $\mathcal{G}^t$ which can best explain their high property scores, and add these discovered substructures as new rationales. Furthermore, we can measure the goodness of these substructure using their mutual information with the property value, and optimize the selector for these substructures using a variational formulation (Chen et al., 2018). This entire procedure is illustrated in Figure 2 can also be seen as seeking explanations why molecules have high property scores.

**Explainer.** A molecular graph $g$ is represented by $g = (\mathcal{V}_g, \mathcal{E}_g)$ with atoms $\mathcal{V}_g$ as vertices and bonds $\mathcal{E}_g$ as edges. For any subset $\mathcal{U} \subseteq \mathcal{V}_g$, the *induced subgraph* $s = (\mathcal{U}, \mathcal{E}_g^{\mathcal{U}})$ is a subgraph of $g$ formed by the vertices $\mathcal{U}$ and the edges $\mathcal{E}_g^{\mathcal{U}} = \{e \in \mathcal{E}_g | e_{start}, e_{end} \in \mathcal{U}\}$ connecting pairs of vertices in the subset. An explainer $\text{Explain}(\cdot) : \mathcal{G} \mapsto \mathcal{S}$ takes a graph $g$ as an input, and outputs an induced subgraph $s$ of $k$ vertices.

**Variational Objective.** We want to learn an explainer for the conditional distribution $\mathbb{P}(Y = 1|g) \triangleq f(g)$ (treating $f(g)$ as a probability), with random variables $Y \in \{0, 1\}$ where $Y = 1$ indicates that the molecule has the property, and 0 otherwise. We will learn a graph vertex sampler $h_\phi(g)$ jointly

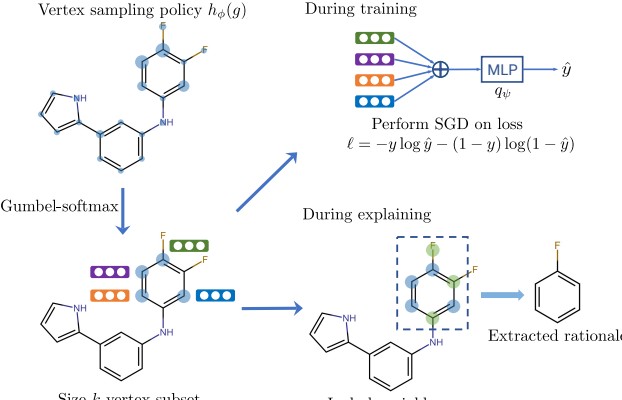

Figure 2: Steps of the explaining process (Alg. 2). The explainer is a subgraph selector containing two steps. First, a vertex sampling policy $h_\phi(g)$ is computed (top-left). Then $k$ vertices are selected using the Gumbel-softmax trick (bottom-left). During training, the embeddings on the selected vertices are pooled together and fed into a MLP $q_\psi$ which predicts the property score (top-right). During explaining, The induced subgraph of the selected vertices and their neighbors is extracted as the predicted rationale.

with a variational approximation $\mathbb{Q}(Y|g)$ of $\mathbb{P}(Y|g)$, such that the mutual information between $Y$ and $s$ is maximized

$$\max_{h_\phi(\cdot),\mathbb{Q}} \mathbb{E}_{Y\sim\mathbb{P}(\cdot|g)}\Big[\log\mathbb{Q}(Y\mid s)\Big], \quad \text{such that } s = (\mathcal{U}, \mathcal{E}_g^{\mathcal{U}}) \text{ and } \mathcal{U} \sim h_\phi(g). \tag{12}$$

Details on sampling $\mathcal{U}$ from $h_\phi(g)$ are presented in the next paragraph. After sampling $\mathcal{U}$, we can construct an induced subgraph $s = (\mathcal{U}, \mathcal{E}_g^{\mathcal{U}})$. During the explanation, we then perform an additional expanding step,

$$s' = (\mathcal{U}', \mathcal{E}_g^{\mathcal{U}'}), \text{ where } \mathcal{U}' = \mathcal{U} \cup \{v|\exists v \in \mathcal{U}, \text{ s.t. } e(u,v) \in \mathcal{E}_g \text{ or } u, v \text{ share a Benzene.}\}, \tag{13}$$

to obtain $s'$, which defines the mapping $s' = \texttt{Explain}(g)$ (Algorithm 2).

**Parameterization of $h_\phi(g)$.** Sampling subgraph $s$ from $g$ is equivalent to sampling a size-$k$ subset $\mathcal{U}$ from the vertices $\mathcal{V}_g$. We use a GNN $h_\phi$ to define a vertex sampling policy $h_\phi(g) \in \Delta_{|\mathcal{V}_g|}$ over the space of $g$'s vertices. Specifically, $h_\phi(g)$ consists of two parts:

1. A message passing network (MPN) which outputs a matrix $X_g = \text{MPN}_\phi(g) \in \mathbb{R}^{|\mathcal{V}_g|\times d}$ representing the $d$-dimensional embeddings for each vertex;
2. A fully-connected layer (FC) followed by a softmax layer to implement the vertex sampling policy $h_\phi(g) = \text{Softmax}(\text{FC}_\phi(X_g)) \in \Delta_{|\mathcal{V}_g|}$.

Then we follow the procedure in L2X (Chen et al., 2018) for sampling $k$ vertices one-by-one from the distribution $h_\phi(g)$ using the Gumbel-softmax trick. The sampled feature matrix can be written as $X_s = V(\phi, \zeta) \odot X_g \in \mathbb{R}^{|\mathcal{V}_g|\times d}$, where $\zeta$ is a collection of auxiliary random variables sampled independently from the Gumbel distribution, $V(\phi, \zeta) \in \{0,1\}^{|\mathcal{V}_g|}$ is a mask on the rows of $X_g$, and $\odot$ is the element-wise product.

**Parameterization of $\mathbb{Q}$.** Since directly using generic choices of $\mathbb{Q}$ to perform the variational approximation is hard, we approximate it with a MLP $q_\psi$ that takes the aggregated masked embedding vector $x_s = \sum_r X_s[r,:] \in \mathbb{R}^d$ as input, and predicts the target via $\mathbb{Q}(Y = 1|s) = q_\psi(X_s) = \text{Sigmoid}(\text{MLP}_\psi(x_s))$.

**Final Objective for Training.** After applying the Gumbel-softmax trick, we transform the variational objective in Eq (12) into:

$$\max_{\phi,\psi} \mathbb{E}_{g,\zeta}\Big[f(g)\log q_\psi(V(\phi,\zeta)\odot X_g) + (1-f(g))\log(1-q_\psi(V(\phi,\zeta)\odot X_g))\Big]. \tag{14}$$

We can then apply stochastic gradient ascent to jointly optimize $\phi$ and $\psi$ by sampling molecule $g$ from the dataset and $\zeta$ from the Gumbel distribution. Please refer to Appendix A.1 for more details of the training procedures as well as the implementation of the explainer.

We note that our design of the explainer model and the learning method is very different from those in GNNExplainer (Ying et al., 2019), which may be of independent interest in terms of explainable models for GNNs. For instance, our explainable model $h_\phi$ by itself is a GNN model, and we introduce a variational distribution $q_\psi$ which is optimized jointly with $h_\phi$.

Table 1: Results on multi-property molecule optimization. `MolEvol` is compared with three variants and four baselines in terms of success rate, novelty, diversity and an overall metric (QNU). The diversity of MSO and GA-D(t) is not reported here due to their extremely low novelty scores.

| Algorithm | MolEvol | [MCTS] | [FixM] | [FixR] | RationaleRL | REINVENT | MSO | GA-D(t) |
|---|---|---|---|---|---|---|---|---|
| Success rate | **93.0%** | 77.7% | 67.3% | 66.3% | 61.1% | 46.6% | 57.7% | 62.0% |
| Novelty | **75.7%** | 72.5% | 67.4% | 54.6% | 57.4% | 66.4% | 28.6% | 19.4% |
| Diversity | 0.681 | 0.707 | 0.723 | 0.727 | **0.749** | 0.666 | - | - |
| QNU | **52.7%** | 47.4% | 39.3% | 28.3% | 29.5% | 7.4% | 16.4% | 12.0% |

**Rationale Extraction as Explaining.** During the E-step in our Algorithm 1, we utilize the trained explainer `Explain(·)` to extract rationales candidates $s$ from the seed molecules. Then the candidates with the top $\mathbb{Q}$-scores are added to the rationale support set to update $\mathcal{S}^t$.

---

**Algorithm 2:** `Explain`$_\phi(g)$

---

**Input:** Molecule $g$, vertex sampling policy $\phi$.
$h_\phi(g) = \text{Softmax}(\text{FC}_\phi(\text{MPN}_\phi(g)))$.
Sample $\mathcal{U} \sim h_\phi(g)$ with Gumbel-softmax trick.
$s' = \text{Expand}((\mathcal{U}, \mathcal{E}_g^{\mathcal{U}}))$ as defined in Eq (13).
**return** $s'$

---

**Remark on Explanation.** In this paper, we use the word "explanation" to refer to a critical component of the input that is of most importance for the final prediction, following the convention of L2X (Chen et al., 2018) and GNNExplainer (Ying et al., 2019). However, a more rigorous explanation using scientific language is rather important and helpful for scientific research. Generating such an explanation using a machine learning model could be highly relevant in general, but that is beyond the scope of this paper.

## 5 EXPERIMENTS

We evaluate `MolEvol` on a multi-property molecule optimization task (Li et al., 2018; Jin et al., 2020) involving four properties:

- GSK-3$\beta$: inhibition levels against glycogen synthase kinase-3 beta (Li et al., 2018);
- JNK3: inhibition levels against c-Jun N-terminal kinase 3 (Li et al., 2018);
- QED: quantitative estimate of drug-likeness (Bickerton et al., 2012);
- SA: synthetic accessibility (Ertl & Schuffenhauer, 2009).

GSK-3$\beta$ and JNK3 are potential targets in the treatment of Alzheimer's disease. Their corresponding property predictors are random forests trained on real-world experimental data using Morgan fingerprint features (Rogers & Hahn, 2010). In our experiment, we consider all properties by combining their scores into a unified scoring function[1]:

$$f(g) = \left[ \text{GSK-3}\beta(g) \cdot \text{JNK3}(g) \cdot \text{QED}(g) \cdot \text{SA}(g) \right]^{\frac{1}{4}}. \tag{15}$$

Note that in the eMolecules dataset (eMolecules, 2020) of commercially available molecules, only 0.007% out of over 27M molecules meet the criteria with $f(g) > 0.5$.

**Experiment Setting.** We provide a set of 3.4K seed molecules for the algorithms to start with. Each seed molecule has a high value in GSK-3$\beta$ or JNK3 or both. There is a budget on both the time and the number of queries. Each algorithm is allowed to query $f$-scores no more than 5M times and to run no more than 1 day on a Ubuntu 16.04.6 LTS server with 1 Nvidia RTX 2080 Ti GPU, and 20 Intel(R) Xeon(R) E5-2678 2.50GHz CPUs. We evaluate the algorithms on 20K generated molecules using the following metrics. We call a molecule $g$ *qualified* if $f(g) > 0.5$, *novel* if the distance between $g$ and the reference molecule set is larger than a threshold[2]. The reference set contains 315 qualified molecules, which is a subset of the provided seed molecules.

- Success rate: the percentage of **qualified** molecules out of 20K molecules.

---

[1]The range of GSK-3$\beta$, JNK3, QED are $[0, 1]$. We re-normalize SA to $[0, 1]$ using $\text{SA}(g) \leftarrow \frac{1}{9}\left(\frac{10}{\text{SA}(g)} - 1\right)$.

[2]$\text{Novel}(g) = \mathbb{I}(\max_{g' \in \mathcal{G}_{ref}} \text{Sim}(g, g') < 0.4)$, $\text{Diversity} = 1 - \frac{2}{n(n-1)} \sum_{g \neq g'} \text{Sim}(g, g')$, $\text{Sim}(\cdot, \cdot)$ is the Tanimoto-similarity on Morgan fingerprints.

- Novelty: the percentage of **novel** molecules out of all **qualified** molecules.
- Diversity: the average pairwise distance between all **qualified** and **novel** molecules[2].
- QNU score: the percentage of **qualified**, **novel** and **unique** molecules out of 20K molecules.

Success rate, novelty and diversity have been adopted as evaluation metrics in previous work (Olivecrona et al., 2017; Li et al., 2018; Jin et al., 2020). However, the trade-off among the three targets complicates the comparisons between algorithms. Therefore we propose a new metric, QNU score, to jointly consider the three aspects. QNU will serve as the major factor for comparison.

**Implementing `MolEvol`.** We first pretrain the graph completion model $p_\theta(g|s)$ on a dataset constructed from ChEMBL (Gaulton et al., 2017), which contains over 1.4M drug-like molecules. The pretraining dataset consists of 4.2M $(s, g)$ tuples, where $g$ is a random molecule from ChEMBL and $s$ is a random subgraph of $g$. In our experiment, `MolEvol` is run for 10 rounds. Within each round, 200 rationales are added to the support set during the explainable local search stage. During the local search stage, 3 to 5 atoms will be sampled according to the vertex sampling policy $h_\phi(g)$ and we include the neighbors of the sampled atoms, *i.e.*, the atoms which share a common bond to the sampled atoms, to form the rationale (Eq (13)). In the molecule completion stage, the parameter $\theta$ is updated with gradient descent for 1 epoch using a total number of 20000 $(s, g)$ pairs with a minibatch size of 10 and a learning rate of 1e-3.

**Baselines.** We compare `MolEvol` against state-of-the-art molecule optimization algorithms below:

- RationaleRL (Jin et al., 2020) learns a graph completion model, but relies on a fixed set of multi-property rationales composed by single-property rationales extracted by MCTS. Concretely, each state in MCTS represents a subgraph of the molecule and the reward function is defined as the property score of the subgraph.
- REINVENT (Olivecrona et al., 2017) learns a RNN model with Reinforcement Learning for generating molecules in the form of SMILES strings;
- MSO (Winter et al., 2019) optimizes the property using Particle Swarm Optimization (PSO) (Kennedy & Eberhart, 1995) in a continuous latent space of molecules.
- GA-D(t) (Nigam et al., 2020) employs a genetic algorithm enhanced with a neural network based discriminator component to promote diversity. The discriminator used here tries to distinguish between molecules generated by the GA and the reference molecule set. The time-dependent adaptive penalty is also used for further promoting exploration.

Since MSO and GA-D(t) do not explicitly learn a generative model, we use the best 20K out of 5M molecules encountered in the search process for comparison.

**Results.** The results are reported in Table 1. Comparing to the baselines, `MolEvol` achieves higher success rate in generating qualified molecules (30% higher than RationaleRL, MSO and GA-D(t), 45% higher than REINVENT). Meanwhile, `MolEvol` maintains high novelty (75.7%) which may benefit from the alternating process in the framework. Although the diversity is slightly lower than RationaleRL due to the distribution shift during optimization, the QNU score, which takes all the above metrics into consideration, is significantly higher than RationaleRL (52.7% versus 29.5%). Please refer to Appendix A.3 for more discussions.

**Ablation Studies.** We introduce baselines below to understand the importance of each component:

- [MCTS] replaces the explainable local search with MCTS as in Jin et al. (2020);
- [FixR] uses a fixed set of rationales, *i.e.* only having one round of explainable local search;
- [FixM] uses a fixed (pretrained) model, *i.e.* having no molecule completion stage.

As illustrated in Table 1, `MolEvol` achieves the highest QNU score among all variants. The large performance gap (success rate: 93.0% vs. 67.3%/66.3%; QNU score: 52.7% vs. 39.3%/28.3%) between `MolEvol` and [FixM]/[FixR] justifies the necessity of taking both E-step and M-step into consideration. Compared with [MCTS], the 5% QNU increase may result from the larger space when doing the local search, while MCTS only proposes connected subgraphs of molecules as rationales.

**Distribution of the Generated Molecules.** In Figure 3-left we plot the evolution of the generative model performance over time. As we can see, the distribution gradually shifts to regions with higher property scores, which demonstrates that `MolEvol` does improve the molecule generative model via EM iteration. As shown in Figure 3-right, `MolEvol` can propose molecules with improved

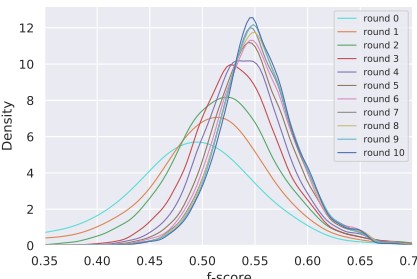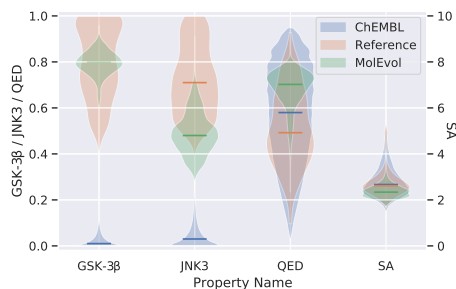

Figure 3: Property score distribution of the generated molecules. Left: the evolution of the $f$-score distribution of MolEvol over the number of iterations. Right: the distribution of four property scores of our generated molecules, the ground truth molecules in the reference set, and the molecules in ChEMBL. The higher the better for JNK3/GSK-3$\beta$/QED, the lower the better for SA.

QED and SA compared to molecules in ChEMBL and the reference set. The distribution for the property scores of molecules generated by MolEvol is more compact than others, which suggests that MolEvol can propose molecules with high property score and low score variance.

**Example of Rationale/Generated Molecule.** Figure 4 gives an example of molecules generated by some rationale discovered using MolEvol. The molecules are of high diversity and pertain consistently high level of scores, which proves MolEvol's superiority.

**Expert Evaluation.** We asked an experienced chemist to evaluate generated molecules. The top-scoring 50 molecules from MolEvol and ChEMBL are selected, shuffled, and grouped with one another to construct 50 pairs. Given a pair of molecules, the chemist is asked to provide a comparative score in each of the four criteria. For the sum of four scores, **30**/50 molecules by MolEvol are higher or identical compared to their counterparts from ChEMBL. For individual scores, **7**/50 molecules by MolEvol are all higher or identical compared to their counterparts. This result shows that our algorithm can propose high-quality realistic molecules that are competitive with existing ones. Please refer to Appendix A.4 for more details.

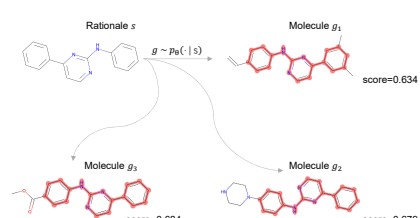

Figure 4: An example of rationale and corresponding generated molecules with $f$-scores.

## 6 DISCUSSION

In this paper, we proposed an EM-like algorithm for optimizing molecules by an explainable evolutionary process. Although we focus our paper and evaluation on molecule design, the method can be generically applied for optimizing discrete structures in other structured prediction domains like program synthesis (Ellis et al., 2020) and graph adversarial learning (Dai et al., 2018a; Zügner et al., 2018). Our method mimics humans' general design process for discrete structures by first identifying useful structural elements and then improving the design based on these elements. The process of discovering more useful substructures and then reiterating the design is carried on to gradually improve the final product. Furthermore, the explainable graph model we developed in the paper can be applied to other general graph problems as well. We believe multiple aspects of our method have broader applications beyond the current molecule optimization problem.

ACKNOWLEDGMENTS

This work is supported in part by NSF grants CDS&E-1900017 D3SC, CCF-1836936 FMitF, IIS-1841351, CAREER IIS-1350983, CNS-1704701, ONR MURI grant to L.S.

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

## A    APPENDIX

### A.1    EXPLAINER IMPLEMENTATION AND TRAINING

Here we provide the details for implementing the graph explainer described in Section 4.3.

For the MPN of $h_\phi$ in the explainer, we use a two-layer GCN with the embedding size of each layer equaling 32. The GCN input is node embedding derived by an Embedding Layer that embeds each node (atom) within the graph (molecule) into a 128-dimensional vector according to its type. The FC layer following MPN outputs a 1-dimensional vector which is used for Gumbel Softmax.

For the MLP of $q_\phi$ in the explainer, we use a two-layer FC network to embed the information, each of whose hidden dimension equals 200. We add a batchnorm layer after each FC layer to make the training phase more stable. After that, a sigmoid layer is used to get the final prediction.

Training procedures are described below.

---

**Algorithm 3:** Training Procedures for the Explainer.

---

**Input:** Molecules dataset $\mathcal{D}$ with each pair $(g, y)$ denoting molecule and label, initial vertex
        sampling policy network $\phi$, MLP network $\psi$ for approximating $\mathbb{Q}$.

1 **for** $t \leftarrow 1$ **to** $N_{epochs}$ **do**
2     Sample $g_1, \cdots, g_m$ from $\mathcal{D}$.
3     **for** $i \leftarrow 1$ **to** $m$ **do**
4         $X_g^i = \text{MPN}_\phi(g_i) \in \mathbb{R}^{|\mathcal{V}_{g_i}| \times d}$.
5         $X_s^i = V(\phi, \zeta) \odot X_g^i \in \mathbb{R}^{|\mathcal{V}_{g_i}| \times d}$, where $\zeta \sim Gumbel(0, 1)$.
6         $\hat{y}_i = q_\psi(X_s^i)$.
7         Update $\phi$ and $\psi$ using gradient ascent by maximizing $f(g_i) \log \hat{y}_i + (1 - f(g_i)) \log(1 - \hat{y}_i)$.

8 **return** $\phi, \psi$

---

### A.2    CONVERGENCE ANALYSIS OF MOLEVOL

From a theoretical standpoint, here we assume 1) we use the true support set $\mathcal{S}$ instead of the finite support set $\mathcal{S}_t$ in (Eq. 8), and 2) $\alpha$ and $m$ in (Eq. 10) are carefully selected such that the gradient update has small enough variance.

**Proof**

- As $J(\theta, p(s))$ has an upper bound, we only need to show that it is non-decreasing over E step and M step.
- E-step: we need to show that $J(\theta^t, p^{t+1}(s)) \geq J(\theta^t, p^t(s))$. It is obvious with assumption 1) as (Eq. 6) has the closed form solution (Eq. 7), so the updated value of $J$ is the maximum after the argmax operation.
- M-step: we need to show that $J(\theta^t, p^t(s)) \geq J(\theta^{t-1}, p^t(s))$. First, it is worth noticing we used the same trick as in REINFORCE to get Eq. 10 from Eq. 9, *i.e.* we can do SGD with the gradient we get in Eq. 10. Then, with assumption 2), by doing SGD, the unbiased gradient estimator with small variance will always converge to a non-decreasing result in the objective value.
- By the above analysis, we can justify that this EM-like method can converge to a local optimum.

                                                                                              ■

Note that both assumptions are rather mild, since for assumption 1), $\mathcal{S}_t$ grows with time $t$ and gradually converges to $\mathcal{S}$, and for assumption 2), a large enough $m$ and a small enough $\alpha$ should suffice.

As will be discussed later, the plot (Figure 5) of the final objective's convergence curve justifies that our algorithm can converge empirically.

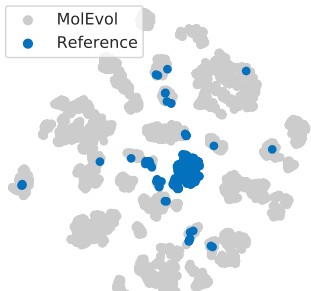 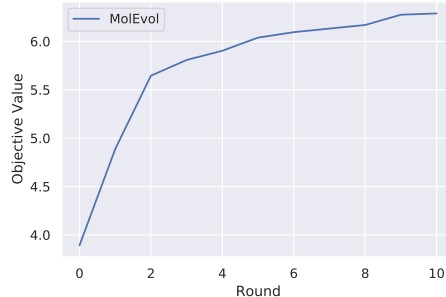

Figure 5: t-SNE plot and the objective value over time. Left: the t-SNE plot of the generated molecules distribution from `MolEvol` and the reference molecules. Right: The diversity-promoting objective (Eq (5)) over time.

### A.3 MORE EXPERIMENT RESULTS AND DISCUSSIONS

**Molecule Distribution.** We projected the generated molecules onto a two-dimensional space by t-SNE (Maaten & Hinton, 2008) together with the reference molecules in Figure 5-left. The molecules generated by `MolEvol` occupy the chemical space expanded by the reference molecules and their neighboring regions.

**Optimization Objective.** We plot the value of $J(\theta)$ in Eq (5) during training. As can be seen in Figure 5-right, the value of objective $J(\theta)$ is consistently improved, which shows that `MolEvol` does help to optimize the diversity-promoting objective in an alternating fashion.

**Analysis of Baselines.** The main reason for MSO's low performance is that it produced molecules with relatively low diversity, so most queries were wasted for evaluating highly similar molecules. Therefore MSO is not well suitable for producing high-scoring molecules with high diversity since there is no regularization for the diversity of molecules it generates. GA-D(t) incorporates the discrimination score to promote unseen molecule generation. However, there is no guarantee that the generated molecules are dissimilar enough to be deemed novel, thus leading to the degradation of overall performance. In comparison with them, REINVENT and RationaleRL resort to REIN-FORCE for optimization, and achieve more competitive performance. Nevertheless, RationaleRL generates molecules from rationales in one-shot, which does not take the insight that the generated molecules might contains some subgraphs (*i.e.* rationales) that are more qualified into consideration.

### A.4 EXPERT EVALUATION EXPERIMENT

We provide more details on the setting of the expert evaluation experiment. We first construct the evaluation molecule set by choosing 50 top-scoring molecules of the same size from our generative model and ChEMBL dataset. The molecules are then grouped into pairs such that each pair contains one from the model and one from the dataset. The order of the two molecules in each pair is randomly shuffled. We then ask experts to evaluate these 50 pairs of molecule with respect to the four molecular properties, *i.e.*, GSK-3$\beta$, JNK-3, QED, SA. For each property, the experts will provide their opinions using one of the following choices:

1. The first molecule is clearly better;
2. The second one is clearly better;
3. The difference is minor and hard to tell.

We use the following two metrics to interpret the result.

- [M-Single]: We score each molecule by summing over the results of all four criteria. A molecule scores 2 points on each criterion if it is clearly better, 1 point if the difference is hard to tell, and 0 points if it is clearly worse. We found that 30 out of 50 generated molecules have better or equivalent scores than its counterpart.

- [M-Overall]: We count the number of pairs where all four properties of the generated molecule are better than or equivalent to the ChEMBL counterpart. Within these pairs, we discard the ones if there is no confident evaluation, *i.e.*, the differences between the pair of molecules on all four criteria are hard to tell. We found that **7** out of 50 remains, meaning that 14% of all the generated molecules are strictly better than their counterpart.

