# OpenReview forum: "Molecule Optimization by Explainable Evolution"
_ICLR.cc/2021/Conference — ICLR 2021 Poster_

### Official Review · AnonReviewer4 · 2020-10-26
**The paper proposes an alternating algorithm to optimise molecule properties based on representative molecule subgraphs. The method is novel and achieves promising results.**

**Rating:** 7
**Confidence:** 4

**Review:**

The paper tackles the problem of molecule property optimisation. To this end, the authors proposes an alternating approach consisting of an explainer model and a molecule completion model. The explainer model takes a complete molecule as input and outputs a subgraph that represents the part that contributes most to property prediction. Then, the molecule completion model uses the subgraphs to sample a complete graph that can maximise the property scores. The loss function of molecule completion model directly maximises the properties, which is non-differentiable so that the authors use a REINFORCE algorithm for optimisation.

Pros:

1. The paper proposes use subgraphs that contributes most to the property prediction for searching better molecules. The subgraphs are learned supervisedly through the signals from the property to be optimised. Compared to unconditional VAE models, this approach might be easier to optimise, since the subgraphs can serve as templates.

2. The method is novel and the experiments demonstrate the effectiveness of the methods compared to previous methods.

3. The paper is well-written and the idea is articulated in a formal description.

Cons (or questions):

1. Some convergence analysis shall be needed, i.e. why this method will converge to the optimal values of the objective. The authors claim the method is an EM algorithm, and some proofs about convergence might be needed. Otherwise, some learning curves might be helpful, since the REINFORCE algorithm is known to suffer from high variances.

---

> ### Author Response · Authors · 2020-11-17
> **Response to Reviewer 4**
>
> Thank you for your constructive comments. We address the convergence of our algorithm below.
>
> This convergence analysis is very similar to EM as we've already defined the components using the terms in the EM framework. But to make it more clear, we give a sketch of the proof below. From a theoretical standpoint, here we assume 1) we use the true support set $\mathcal{S}$ instead of the finite support set $\mathcal{S}_t$ in (Eq. 8), and 2) $\alpha$ and $m$ in (Eq. 10) are carefully selected such that the gradient update has small enough variance.
>
> **Proof:**
> - As $J(\theta, p(s))$ has an upper bound, we only need to show that it is non-decreasing over E step and M step.
> - E-step: we need to show that $J(\theta^{t}, p^{t+1}(s))\geq J(\theta^{t}, p^{t}(s))$. It is obvious with assumption 1) as (Eq. 6) has the closed form solution (Eq. 7), so the updated value of $J$ is the maximum after the argmax operation.
> - M-step: we need to show that $J(\theta^{t}, p^{t}(s))\geq J(\theta^{t-1}, p^{t}(s))$. First, it is worth noticing we used the same trick as in REINFORCE to derive Eq. 10 from Eq. 9, i.e. we can do SGD by the gradient we get in Eq. 10. Then, with assumption 2), by doing SGD, the unbiased gradient estimator with small variance will always converge to a non-decreasing result in the objective value.
> - By the above analysis, we can justify that this EM-like method can converge to some local optimum.
>
> We further note that both assumptions are rather mild, since for assumption 1), $\mathcal{S}_t$ grows with time $t$ and gradually converges to $\mathcal{S}$, and for assumption 2), a large enough $m$ and a small enough $\alpha$ should suffice.
>
> Also, we’d like to mention that in the appendix, we have a plot (Fig. 5) of the final objective’s convergence curve to justify that empirically our algorithm can converge.

---

### Official Review · AnonReviewer2 · 2020-10-28
**Nice idea, but structure & clarity could be better**

**Rating:** 6
**Confidence:** 3

**Review:**

The authors propose a two step procedure for generating molecular graphs that optimize some desirable properties. The method consists of a rationale extraction phase, where the subgraph "important" for the desired property is identified and a graph completion step.

When reading the paper for the first time, I found it a bit hard to follow the approach. The paper might be easier to read when the individual model components are introduced directly with the E- and M-step. Currently, the graph completion model is introduced in Sec 3.1, then the E- and M-steps are described in Sec. 3.2, while the "explainer" is only mentioned and referred to Sec. 3.3. An easier to follow structure might be: 1) Rationale extraction 2) Graph completion 3) E/M iteration.
Beyond that, the terms "evolution" and "explanation" might be misleading here: The method is not an evolutionary algorithm, as the title might suggest. Even if subgraphs are extracted that lead to the generation of promising molecules in the graph completion step, this does not show that these subgraphs responsible. To warrant the term explaination, a  thorough analysis of the extracted rationales would be necessary, in particular since they might not even be connected graphs.

The evaluation of the method in Table 1 and Fig. 3 show that the method outperforms previous appraoches and is capable of shifting the generated distribution shifts to higher scores. The human evaluation is weak, since only a single expert was asked. A panel of experts with reported agreement among the panel would improve the paper. However, since the scores of the molecules are available, it does not become quite clear to me, how human experts benefit the performance evaluation under those same criteria.

Pros
-----
- Interesting approach to update the pool of rationales
- Outperforms previous approaches, Fig 3. show that desired properties improve over rounds
- Ablation study demonstrates improvement by proposed procedure

Cons
----
- Structure of the paper could be improved
- The paper states that in the explainer a "subgraph s of k vertices" is extracted, therefore I assume the size of the rationale is fixed. This would severely restrict the space of rationales.
- The notion of explainability is not sufficiently discussed in the paper and the claim(?) that the rationales are somehow meaningful is not examined.

Update: I read the reply and thank the authors for the clarifications.

---

> ### Author Response · Authors · 2020-11-17
> **Response to Reviewer 2**
>
> Thank you for your constructive comments. We address your concerns below in detail.
>
> **1. Structure of the Paper.**
>
> Thanks for bringing this up. We are aware of the issues in the presentation and will improve the writing and organization in the next revised version.
>
> **2. The Space of Rationale.**
>
> In fact, k is not fixed during our experiment, we randomly sample k from 3 to 5 for selecting atoms during the local search phase and during training. We will add these details into our revised version. Furthermore, each rationale is generated by additional expansion from the selected atoms to complete the partial bonds (Eq. 13), which could lead to a flexible enough search space for finding good rationales.
>
> **3. Notion of Explainability.**
>
> We’re sorry about the lack of preciseness for the notion of explainability. We were intended to use the word "explainability'' to describe the ability of a subgraph to generate high scoring molecules, rather than using chemical science to explain what exactly causes the high scores. Also we showed the results in Fig. 4 to explain that a good rationale can be a good inductive bias in designing good molecules (as we use $p_\theta(g|s)$ to decode some good molecules by the rationale). We will address these issues (and the potentially misleading concept of "evolutionary'') in our revision.
>
> We totally agree with you and R3 that a more rigorous explanation using scientific language is important and helpful for the material scientists. This explanation itself could be very interesting in general even without the context of molecule optimization. We would put that in the future work.

---

### Official Review · AnonReviewer1 · 2020-10-28
**novel approach for molecular design using explainability**

**Rating:** 7
**Confidence:** 5

**Review:**

### Initial Review

The paper proposes a new algorithm for de-novo molecular design, which uses a model to extract explanatory subgraphs from a set of support molecules which "explain" high scores wrt a scoring function, and a generative model which is conditioned on these subgraphs to produce full molecules.

All in all, I think this is an interesting combination of several existing approaches in generative models for molecules (all referenced in the paper). The aspect of explainability is novel. The presentation of the paper is mostly clear. The approach is quite geared to molecule generation, but can potentially also inspire applications in other domains, which makes it interesting from a general ML perspective as well.

I like the paper from the theoretical side, which alone warrants acceptance of the paper at ICLR in my opinion.

However, I have a few minor concerns / comments:

I am a bit on the fence with the validation method. Since almost every new paper in the field proposes a new validation approach, it has become pretty much impossible to assess what the state of the art of the field is (or if the concept of SOTA is even something meaningful), and this paper is no exception in this regard. But I assume the authors will disagree here.

In practice, generating 20k molecules is a lot. Looking at the statistics of the top100 molecules would probably be sufficient.

Also, I find it somewhat surprising that some of the baseline algorithms (in particular the Winter et al MSO model), which are less constrained than algorithm presented here, are not achieving higher scores, in particular when the algorithms can query the scoring function 5 M times. Maybe this is something the authors could comment on in the rebuttal.

As an additional baseline, I would suggest to report the "best in dataset", I.e. run the scoring function on the seeds and all molecules used to train the generator, and pick the top molecules.


Related work:

I would suggest to additionally cite https://arxiv.org/abs/1701.01329 which was the first paper to apply neural models to molecule generation in drug discovery, and the first of such papers which has been prospectively validated in laboratory experiments by scientists not affiliated with the authors.

### Update 1 after discussion:
Score raised.

---

> ### Author Response · Authors · 2020-11-17
> **Response to Reviewer 1**
>
> Thank you for your constructive comments. We address your concerns below in detail.
>
> **1. Evaluation Metric.**
>
> The evaluation metric we used in the paper is mostly similar with RationaleRL ([1]), i.e. we compare the success rate, novelty and diversity here. Moreover, we hope to use a more comprehensive metric that can jointly take these three factors into consideration, so we propose QNU as an overall metric, as explained in the 3rd paragraph on page 7.
>
> For the reason why we used 20k molecules for evaluation, the goal of our work is that to learn a sampler being able to generate some diverse and novel molecules with high property value over a highly discrete space (also of high variance), i.e. we want to learn a distribution instead of a few points with high value, which tends to be a more general problem. In fact, the setting for evaluation is close to what RationaleRL used, by decoding each rationale with a fixed number of times. Furthermore, if we only evaluate the molecules with highest values, it is very likely that these molecules are similar to each other (and to the seed molecules) and of low diversity, which is meaningless for helping scientists to design novel drugs.
>
> **2. Low performance in the MSO baseline.**
>
> We ran MSO with a swarm size of 20000 for 250 epochs using our defined score function f(g) (in Eq. 15). The main reason for MSO’s low performance is that it produced molecules with relatively low diversity, so most queries were wasted for evaluating the same molecules. In fact, we ran MSO for more than 1 day to get the reported result, which actually exceeds the resource budget in our experiment setting. We conjecture that MSO is not well suitable for producing high scoring molecules with high diversity since there’s no regularization for the diversity of molecules it generates.
>
> **3. Best in Dataset.**
>
> The statistics of all the scores for the generated molecules and seed molecules: the best score in the seed molecules is 0.713, while ours is 0.716. However, we’d like to address that only comparing the best might not meaningful since our method is designed to generate a series of molecules all achieving a good score. Therefore, "best in dataset“ could not reflect the performance gain of the ``“distribution learning” by our method, and so we didn't treat it as a proper metric in the paper.
>
> [1] Jin, Wengong, et al. "Multi-Objective Molecule Generation using Interpretable Substructures" in ICML 2020.

---

> > ### Comment · AnonReviewer1 · 2020-11-25
> > **thanks for the clarification**
> >
> > Thanks for the clarification and additional discussion, that is quite insightful. I've increased my rating to reflect that.
> >
> > I think the inclusion of the best in dataset baseline would still be very instructive (or rather the mean of topK in dataset to reflect your evaluation setup more) for practitioners, because best in dataset somewhat reflects the virtual screening setup people would use in practice.
> >
> > as a suggestion for future work:
> > It would be good to additionally run the Guacamol benchmarks in the future, because these are widely accepted, and many baseline results are already available (see e.g. two papers at this year's NeurIPS, but also MSO and REINVENT)

---

### Official Review · AnonReviewer3 · 2020-10-30
**Review of Molecule Optimization by Explainable Evolution**

**Rating:** 8
**Confidence:** 4

**Review:**

SUMMARY:

The authors propose a method that "explains" molecular properties based on molecular fragments and call it Molecular Evolution. The subgraph "explanations" then are used to explore larger swaths of chemical space.

PROS:

- As far as the reviewer notes, this approach is novel in the (now increasingly crowded) set of alternatives for molecular generative models.
- The authors have a model that compares favorably to the baselines
- The authors use a very relevant set of optimization parameters for the multiobjective task.
- The paper is well explained.


CONS:
- The reviewer believes that there is much more to explain why a molecule is better for a task than identifying a subgraph. This should be made clear in the manuscript as materials scientists want to know for example quantum properties of the fragment(s) and how they influence the given property to provide a valuable explanation.

---

> ### Author Response · Authors · 2020-11-17
> **Response to Reviewer 3**
>
> Thank you for your constructive comments. We address the explainability issue below.
>
> We are aware that the lack of preciseness for the notion of explainability might cause confusion in our work. We used the word to describe the ability of a subgraph to generate high scoring molecules, as demonstrated in our paper. We will address this issue in our revision.
>
> However, we think the reviewer raises an important point. Using scientific terms/measurements for explanation is valuable and would greatly benefit scientific research. In order to achieve the goal, we might need to incorporate more scientific knowledge as inductive bias into our explanation model to support a more rigorous explanation. Currently, our method cannot provide such a guarantee and we would leave that in future work.

---

### Author Response · Authors · 2020-11-24
**Revision Uploaded**

We sincerely thank all the reviewers for their constructive comments. We have revised our paper accordingly with the promised explanation and implementation details included. Please check out the new version!

1. We updated the explainable graph model section (Section 4.3, Remark on Explanation) with the clarification for "explanation".

2. In Appendix A.3, We included implementation details of rationale sampling during the local search stage.

3. In Appendix A.4, We added a discussion on the convergence of the proposed algorithm.

4. We updated the experiment section (Appendix A.5, Analysis of Baselines) with more discussions on the performance of baselines such as MSO.

We will keep improving the writing and organization of the paper. If there are any additional comments on the paper, please don’t hesitate to let us know.

---

### Decision · Program_Chairs · 2021-01-07
**Final Decision**

**Decision:**

Accept (Poster)

**Comment:**

The authors appreciated this submission because (a) the aspect of explainability is novel, (b) its strong performance, (c) the clarity of the paper. I urge the authors to double check all of the reviewer comments to make sure they are all addressed in the updated version. I vote to accept.